# tRF-BERT: A transformative approach to aspect-based sentiment analysis in the bengali language

**Shihab Ahmed**[1], **Moythry Manir Samia**[1], **Maksuda Haider Sayma**[2], **Md. Mohsin Kabir**[3,4], **M. F. Mridha**[5]*

1 Department of Information and Communication Technology, Comilla University, Cumilla, Bangladesh, 2 Department of Computer Science and Engineering, CCN University of Science & Technology, Cumilla, Bangladesh, 3 Department of Computer Science and Engineering, Bangladesh University of Business and Technology, Dhaka, Bangladesh, 4 Faculty of Informatics, Eötvös Loránd University, Budapest, Hungary, 5 Department of Computer Science, American International University-Bangladesh, Dhaka, Bangladesh

* firoz.mridha@aiub.edu

**Data Availability Statement:** The datasets supporting this study are openly available in the "Bangla_ABSA_Datasets" repository on GitHub. The

## Abstract

In recent years, the surge in reviews and comments on newspapers and social media has made sentiment analysis a focal point of interest for researchers. Sentiment analysis is also gaining popularity in the Bengali language. However, Aspect-Based Sentiment Analysis is considered a difficult task in the Bengali language due to the shortage of perfectly labeled datasets and the complex variations in the Bengali language. This study used two open-source benchmark datasets of the Bengali language, Cricket, and Restaurant, for our Aspect-Based Sentiment Analysis task. The original work was based on the Random Forest, Support Vector Machine, K-Nearest Neighbors, and Convolutional Neural Network models. In this work, we used the Bidirectional Encoder Representations from Transformers, the Robustly Optimized BERT Approach, and our proposed hybrid transformative Random Forest and Bidirectional Encoder Representations from Transformers (tRF-BERT) models to compare the results with the existing work. After comparing the results, we can clearly see that all the models used in our work achieved better results than any of the previous works on the same dataset. Amongst them, our proposed transformative Random Forest and Bidirectional Encoder Representations from Transformers achieved the highest F1 score and accuracy. The accuracy and F1 score of aspect detection for the Cricket dataset were 0.89 and 0.85, respectively, and for the Restaurant dataset were 0.92 and 0.89 respectively.

## Introduction

In recent times, the expansion of digital media, particularly with the rise of social media platforms, has stimulated an information revolution. This surge in information has attracted diverse researchers interested in understanding various facets of human behavior, including their opinions and sentiments on different subjects. The collection of such data, if performed

datasets can be accessed directly via https://github.com/atik-05/Bangla_ABSA_Datasets.

**Funding:** The author(s) received no specific funding for this work.

**Competing interests:** The authors have declared that no competing interests exist.

effectively, can offer insights into people's circumstances, expectations, and health, encompassing both physical and psychological aspects.

Researchers have long been intrigued by how individuals express themselves during significant events such as global pandemics, natural calamities, or socio-political disruptions. For instance, during the COVID-19 pandemic, social media platforms played a vital role in facilitating communication between governments, emergency responders, and the public amidst social distancing measures. Nowadays, through sentiment analysis, valuable patterns concerning various issues impacting public life can be gleaned from online platforms. Sentiment analysis, a process leveraging natural language processing (NLP) techniques, is instrumental in discerning and categorizing the various emotional states (positive, negative, or neutral) expressed by individuals or communities regarding specific topics [1]. This analytical tool has applications across diverse fields such as business, marketing, and medical science. For instance, in a study conducted by Quin Xiang Ng et al., they explored the prevalence of negative sentiments surrounding flu vaccination, particularly fueled by misinformation circulating on social media platforms like Twitter. Their findings underscored the significant adverse effects of the COVID-19 pandemic and the associated misinformation on public perceptions of flu vaccination, necessitating strategies to address trust issues and combat misinformation to enhance vaccination rates [2]. Similarly, researchers delved into negative sentiments surrounding measles vaccination using Twitter data in another study, further highlighting the implications of social media discourse on public health initiatives [3].

A natural language processing (NLP) technique called aspect-based sentiment analysis (ABSA) determines one's perspective of the specific features of an entity in a topic. ABSA differs from conventional sentiment analysis. Traditional sentiment analysis simply determines whether a sentence is positive, negative, or neutral rather than analyzing the individual aspects of the discussed topic. However, in ABSA, both aspect and sentiment are considered and the sentiment polarity is identified for each aspect of a topic, service, or product [4,5]. For example, "The performance of this laptop is top-notch, but it's too expensive for me to justify buying". Here, ABSA considers the aspects of "performance" and "price" while determining the sentiment. The review highlights the positive aspects of product performance but expresses the negative aspects of the product price. The popularity of ABSA has seen an increased demand for analyzing text, especially for analyzing customer satisfaction, public perception patterns, social media post analysis, product review analysis, and so on.

ABSA offers the ability to break down and understand different types of opinions in a structured way. Therefore, it has been used in different languages worldwide, including English, Hindi, and Arabic, for analyzing texts and making decisions based on the results. Researchers have developed multiple ABSA datasets, including the Twitter dataset [6] and the SemEval-2014 ABSA Task dataset [7], which are mainly based on restaurant reviews and laptop reviews. Various neural network architectures, including the CNN with attention mechanism [8,9] and Graph Convolutional Network (GCN) [10], have been successfully employed for aspect-based sentiment classification tasks. [11] also proposed a GCN-based model to enhance the Bidirectional Long Short-Term Memory (BiLSTM) model's feature representation. Various machine learning techniques, including Random Forest (RF), Decision Tree (DT), Extra Tree Classifier (ET), and support vector machine (SVM), along with deep learning approaches like Gated Recurrent Unit (GRU) and BiLSTM, have been employed in [12]. Other methods like dual-transformer neural network architecture [13] and bidirectional encoder representations from transformers (BERT) [14] are also popular for ABSA tasks. In addition, a better version of the GCN method named Syntactic and Semantic Enhanced GCN (SSEGCN) is used in [15] for aspect-based sentiment classification.

With its extensive historical and linguistic legacy, Bengali is no exception, and it is becoming more and more important for organizations, researchers, and other entities to understand the complex expressions in this language. However, ABSA is challenging for the Bengali language, as it has a rich vocabulary and syntax. This is why it is sometimes difficult to identify the exact aspect and extract the sentiment from Bengali text.

Another challenge is that few resources are available in the Bengali language for NLP research. This means that the Bengali language lacks many labelled datasets, making it more challenging to develop new ABSA models and evaluate and compare different ABSA models. Despite these hardships, Rahman *et al.* created the first Bengali ABSA dataset [16] based on cricket and restaurant reviews. The BAN-ABSA [17] dataset is another popular Bengali dataset. Several models have already been developed for the Bengali ABSA task, such as a CNN-based architecture [18], the three-stacked auto-encoders model [19], Deep-ABSA based on character-embedded convolutional neural network (CNN), and Bi-LSTM with attention mechanism [20]. A variety of machine learning techniques, including Bernoulli Naive Bayes (NB), Multinomial Naive Bayes (MNB), Linear Regression (LR), SVM, Stochastic Gradient Descent (SGD) classifier, Multi-layer Perceptron (MLP) classifier, RF, etc. are used in [21, 22].

This study aims to create a foundation for ABSA in the Bengali language and overcome the challenges in developing and evaluating more accurate and robust ABSA models. To accomplish this goal, we used advanced NLP approaches to assess the distinctive linguistic features of the Bengali language effectively. A crucial part of this research involves identifying a dataset specifically designed for Bengali aspect-level sentiment analysis. For this research, we used the publicly available 'Cricket' and 'Restaurant' datasets created by Rahman *et al.* [16], considered the benchmark datasets for Bengali ABSA. Each entry in the dataset goes through a comprehensive labelling process that produces sentiment labels for various aspects.

This study leverages modern transformer-based NLP techniques, such as BERT and the robustly optimized BERT approach (RoBERTa), which have exhibited exceptional performance across various NLP tasks. However, their performance on English ABSA datasets is remarkable, while adapting them to the intricacies of the Bengali language presents unique challenges. To address this, we meticulously fine-tuned these already pre-trained models using the Bengali ABSA dataset, involving nuanced parameter adjustments to optimize their effectiveness in comprehending and analyzing sentiment nuances expressed in Bengali text.

Additionally, our research introduces a novel hybrid model, transformative Random Forest and Bidirectional Encoder Representations from Transformers (tRF-BERT), which leverages the strengths of both BERT and RF to enhance aspect classification accuracy. Hybrid models are attracting the attention of researchers as they facilitate the leveraging of the strengths of various techniques. Lin *et al.* [23] introduced an approach that utilizes a combination of BERT [14] and a distilled version of BERT, called DistilBERT, to represent input sentences. Furthermore, they proposed a hybrid deep learning classification method for sentiment analysis in the Indonesian language, which integrates BiLSTM and Temporal Convolutional Networks (TCN).BERT [14], an advanced deep learning model, excels in capturing intricate linguistic patterns and capitalizes on transfer learning. On the other hand, RF improves the model's interpretability, resilience, and aptitude for various perspectives. These strategies consider the specific grammatical nuances of the Bengali language, resulting in substantial improvements in precision and recall for aspect identification. By striking a careful balance between model complexity and empirical validation, our hybrid approach surpasses standalone BERT [14] and RoBERTa [24] models in aspect classification, underscoring its practical utility in real-world applications. Our hybrid model outperforms existing methods in two publicly available Bengali datasets, underscoring its effectiveness across diverse linguistic contexts.

The following describes our research efforts in the field of Bengali ABSA:

- We introduced an innovative hybrid model, tRF-BERT, which combines BERT [14] and RF, tailored for Bengali sentiment analysis. This approach excels in handling the complexities of the Bengali language and surpasses the existing performance models.

- In addition to the hybrid models, we utilize cutting-edge language models, RoBERTa [24] and BERT [14], diversifying our approach and improving research versatility.

- Notably, our work achieves the highest performance on two publicly available Bengali ABSA datasets, marking a significant advancement in the field. Our models exhibit exceptional proficiency in analyzing sentiment in Bengali text, which is a less-explored linguistic domain.

- Our study contributes significantly to the underexplored domain of ABSA in the Bengali language by addressing a notable gap. It enhances the applicability of sentiment analysis in Bengali, benefiting from tasks such as customer feedback and product review analysis in the Bengali-speaking context.

The rest of the paper is structured as follows: the Literature review section provides an extensive overview of relevant literature in the field of ABSA. The Problem statement section clearly discusses the overall problem statement and the challenges it presents. The Methods and materials section explains the methodology for our proposed hybrid tRF-BERT model, data collection, and preprocessing. The Result analysis section presents the results of our comprehensive experimental, comparing our model's performance against existing methods in the field of Bengali ABSA. The Discussion section discusses why our model performs better than other models. Finally, the paper concludes with a detailed summary of our work's key findings and contributions in the Conclusions section.

## Literature review

ABSA is considered a classification challenge as it is a more precise form of sentiment analysis. Liu, B. first introduced the concept of the ABSA in his research paper and discussed its methods [25]. The benchmark datasets for ABSA were developed and published by Pontiki *et al.* [7]. They created datasets using restaurant and laptop reviews and labelled them manually. They divided their whole work into four subtasks. For the first two subtasks (aspect term extraction, aspect term polarity) they used both restaurant and laptop datasets, and for the last two subtasks (aspect category detection, aspect category polarity) they used only restaurant review datasets. Karimi *et al.* presented a distinct adversarial training architecture in ABSA, surpassing both the general-purpose BERT model and the BERT model fine-tuned for specific domains in terms of performance, in both feature extraction and sentiment classification tasks [26]. This demonstrates how adverse situations can improve the effectiveness of BERT models during training on networks. By combining dense neural networks, dependency parsing, and POS tagging, Suciati *et al.* suggested an ensemble method for aspect extraction [12]. They discovered that this strategy outperformed lexicon-based strategies, highlighting the advantages of combining deep learning with traditional techniques. In their study on ABSA context modelling, Xing *et al.* recognized and addressed the aspect-agnostic problem [27]. Their work emphasized the importance of treating the semantics of the given aspect as a new and distinct piece of information, separate from the surrounding context itself. To address this issue, they proposed aspect-aware context encoders, such as aspect-aware BERTs (AABERTs), aspect-aware GCN (AAGCN), and aspect-aware LSTM (AALSTM), which can provide concealed states that are aspect-aware and designed explicitly for the ABSA challenge.

ABSA is also gaining popularity for analyzing Bengali datasets. This is because there is an increasing amount of Bengali text data available, and Bengali is becoming more significant in the global economy. Rahman *et al.* introduced ABSA to the Bengali language by creating two free benchmark datasets: 'Cricket' and 'Restaurant' [16]. The 'Cricket' dataset comprises 2900 manually labelled cricket comments, meticulously categorized into five distinct aspect categories. Similarly, the 'Restaurant' dataset encompasses 2600 customer reviews for in-depth analysis. They also developed a CNN-based model for aspect category extraction in the same year that they created their dataset [18]. Their developed CNN model outperformed some well-known machine-learning algorithms on their datasets [16], achieving an F1-Score of 51% on 'Cricket' and 64% on 'Restaurant'. For machine-learning algorithms, SVM, KNN, and RF categorization, they obtained an F1-Score of 35%, 34%, and 37% respectively on the 'Cricket' dataset, and 42%, 38%, and 38% on the 'Restaurant' dataset. Using the datasets of Rahman *et al.* [16], Bodini developed a three-stacked Auto-encoders (AE) model for aspect categorization in Bengali text [19]. Stacked AEs are similar to a team of neural networks that are trained layer by layer. Each layer of a stacked AE learns to encode the previous layer's output. The training process involved stacking three layers: AE, sparse AE (SAE), and contractive AE (CAE).

Boidini's models demonstrated superior performance to those of Rahman *et al.* [16, 18] across all metrics, encompassing precision, recall, and F1 score. Notably, the CAE model achieved the highest F1 score on the "Restaurant" and "Cricket" datasets, with scores of 0.87 and 0.91, respectively. Haque *et al.* developed a new model for Bengali ABSA using a traditional machine-learning method with minimal data preprocessing [28]. They used the benchmark datasets developed by Rahman *et al.* [16]. They found that it is possible to achieve a higher F1-Score if they preprocessed the data less. Priority Sentence Part Weight Assignment (PSPWA), a novel method for extracting features from Bengali text, was proposed by F. A. Naim [29] and is based on the datasets contributed by Rahman *et al.* [16]. The PSPWA was evaluated using two machine learning methods: CNN and traditional supervised learning. The CNN outperformed traditional supervised machine learning methods on both the "Cricket" and "Restaurant" datasets. The CNN model achieved an F1-score of 0.59 for the "Cricket" dataset. For the "Restaurant" dataset, the CNN model's F1-score was 0.67, demonstrating its improved accuracy in classifying sentiment-aspect pairs for restaurant reviews.

Hridoy *et al.* investigated the sentiment of Bangla newspaper headlines based on specific aspects [21]. They employed the ABSA techniques. Their analysis used a relatively small training dataset for various classifiers, including Bernoulli Naive Bayes, Multinomial Naive Bayes, LR, SVM, SGD, MLP, RF, and Support Vector Classifier (SVC). During their experimentation, it was observed that Bernoulli Naive Bayes outperformed the other classifiers, achieving an impressive F1 score of 70.75.

Sultana *et al.* introduced a model for Bengali ABSA by utilizing the Bengali NLP [22]. Their dataset comprised 4012 Bengali text comments on music, drama, movies, and cricket, sourced from YouTube. They employed well-established supervised machine learning methods, including SVC, RF, and LR. Their results demonstrated an accuracy of over 75% when categorizing sentiments as neutral, positive, or negative.

Additionally, the model exhibited an 80% accuracy rate for successfully identifying aspects within Bengali text.

Deep-ABSA is a novel deep-learning framework created by Islam *et al.* for the ABSA of Bengali text [20]. Their framework adopted a multi-channel design, integrating diverse elements. In one channel, they implemented word embedding, Bi-LSTM, and an attention mechanism. In another channel, they utilized CNN with character embedding. Eventually, these two channels are combined by merging their respective features. To evaluate their model, they applied it to the BAN-ABSA dataset, achieving an accuracy of 81% and F1 score of 82%.

Zhang *et al.* introduced SSEGCN, a novel model for ABSA [15]. Their proposed mechanism combined aspect-aware attention and self-attention. This mechanism can learn both aspect-related semantic correlations and the sentence's overall meaning. They evaluated their model on three widely used benchmark datasets: SemEval-2014 Task 4 [7] for restaurant and laptop reviews and Twitter sentiment classification from [6]. The SSEGCN achieved state-of-the-art performance on all three datasets. The accuracies for the restaurant, laptop, and Twitter datasets were 84.72%, 79.43%, and 76.51% respectively. They further improved the accuracy to 87.31%, 81.01%, and 77.40% respectively by combining BERT [14] with their SSEGCN model.

Tang *et al.* introduced the Dependency Graph Enhanced Dual-Transformer Network (DGEDT), which is a dual-transformer neural network architecture designed explicitly for aspect-based sentiment classification [13]. Their approach aims to overcome the limitations of existing methods by combining the strengths of flat sentence representations and graph-based sentence representations. The authors conducted their research on five datasets, including the Twitter dataset from [6], the Lap14 and Rest14 datasets from SemEval-2014 Task 4 [7], the Rest15 from SemEval-2015 Task 12 [30], and the Rest16 dataset from SemEval-2016 Task 5 [31]. They achieved an overall accuracy of 74.8%, 76.8%, 83.9%, 82.1%, and 90.8% for these datasets using their DGEDT model, which further increased to 77.9%, 79.8%, 86.3%, 84%, and 91.9% respectively by combining BERT with their proposed DGEDT model.

The overview of this literature review is displayed in Table 1.

**Table 1. Overview of literature review.**

| Research | Year | Dataset | Used Model | Outcome |
|---|---|---|---|---|
| Suciati *et al.* [12] | 2020 | from PergiKuliner platform | SVM, LR, DT, ET | Food: 88.16%, Price: 89.54%, Service: 89.03% Ambience: 84.78% |
| Xing *et al.* [27] | 2022 | Lap14, Rest14 [7] | (Bi-)AALSTM, AAGCN, AABERT | Lap (t), Rest (e): 73.13 Rest (t), Lap (e): 75.69 t: trained, e: evaluated |
| Pontiki *et al.* [7] | 2014 | Restaurant and laptop sectors reviews | Conditional random fields (CRF) with features extracted | Laptop: 74.55% Restaurant: 84.01% |
| Karimi *et al.* [26] | 2021 | Rest16 [32], Lap14, Rest14 [7] | BERT | Laptop: 85.57% Restaurant: 81.50% |
| Bodini *et al.* [19] | 2019 | Cricket Restaurant [16] | AE, SAE, CAE | Cricket: 0.88 Restaurant: 0.87 |
| Naim *et al.* [29] | 2021 | Cricket Restaurant [16] | CNN, RF, SVM etc. | Cricket: 0.59, Restaurant: 0.67 |
| Haque *et al.* [28] | 2020 | Cricket Restaurant [16] | SVM, RF, LR etc. | Cricket: 0.37, Restaurant: 0.43 |
| Rahman *et al.* [18] | 2018 | Cricket Restaurant [16] | CNN, RF, SVM etc. | Cricket: 0.51, Restaurant: 0.64 |
| Hridoy *et al.* [21] | 2021 | Bangla newspaper headlines | RF, Multinomial & Bernoulli NB, LR, SVM, MLP, SGD & Voting Classifier | Bernoulli NB: 70.75 |
| Sultana *et al.* [22] | 2022 | 4012 Bangla youtube comments | SVC, RF, LR | aspect: 80% sentiment: 75% |
| Islam *et al.* [20] | 2023 | BAN-ABSA t[17] | BiLSTM CNN | 82% |
| Zhang *et al.* [15] | 2022 | Lap14, Rest14 [7] Twitter sentiment dataset | SSEGCN+BERT, SSEGCN, DGEDT+BERT, BERT, T-GCN+BERT, Bidirectional GCN, DGEDT, etc. | 87.31% |
| Tang *et al.* [13] | 2020 | Lap14, Rest14 [7] Rest15 [30], Rest16 [31], Twitter dataset | DGEDT-BERT, TG-BERT, DGEDT, BERT, LSTM, Capsule Network (CAPSNet), etc. | 91.9% |

## Problem statement

This research was motivated by the fundamental goal of creating an advanced ABSA system tailored to the Bengali language. This system goes beyond merely identifying the overall sentiment in Bengali text; it dives into the intricate task of recognizing sentiment associated with specific aspects or features mentioned within the text. To achieve this, a novel ABSA methodology, named "tRF-BERT", synergizes the power of a BERT-based and RF models.

Traditional sentiment analysis categorizes text into broad positive, negative, or neutral classes. ABSA, on the other hand, aims to dissect and attribute sentiment to specific elements within the text. For example, in a sentence such as "The laptop's performance is stellar, yet its price is high", ABSA seeks to distinguish the positive sentiment regarding "performance" and the negative sentiment regarding "price".

This research focuses on the Bengali ABSA, which is relatively unexplored. Notably, it introduces transformer-based models such as BERT [14] and RoBERTa [24], which were lacking in previous Bengali ABSA studies. It also addresses the need for improved accuracy in existing models.

One distinctive feature of this study is the introduction of a hybrid model, tRF-BERT, which combines machine learning and deep learning. This novel approach is expected to enhance ABSA's robustness and accuracy in Bengali.

The chosen hybrid approach, combining BERT and Random Forest models, is grounded in the intention to capitalize on the unique strengths of each model in addressing challenges specific to Aspect-Based Sentiment Analysis (ABSA) in Bengali. BERT, renowned for its ability to capture contextual nuances and intricate linguistic patterns, is well-suited to comprehend sentiment dependencies in text. This particularly benefits ABSA, where understanding sentiments toward specific aspects or entities is paramount. On the other hand, the Random Forest model provides interpretability and stability, acting as a transparent baseline for classification tasks. The hybrid model strives to synergize these advantages by leveraging BERT's deep contextual insights alongside Random Forest's interpretability, thereby enhancing the overall model performance. This hybrid approach offers a balanced solution in the context of ABSA challenges, such as aspect extraction, limited data, and the need for model interpretability. BERT assists aspect extraction, while Random Forest's stability contributes to reliable predictions in scenarios with constrained data. Additionally, the ensemble effect aims to improve model generalization, addressing the complexities of sentiment analysis in Bengali by providing a comprehensive and efficient solution.

A rigorous cross-validation strategy involving the RoBERTa and BERT models is adopted to ensure research credibility. This approach strengthens the trustworthiness of the findings and facilitates a comprehensive exploration of ABSA in Bengali, ultimately advancing the field and expanding its practical applications.

To formalize the problem, let $X_i$ represent different components such as tRF-BERT, Transformer Models, Cross Validation Strategy, and Accuracy Improved, and $w_i$ represent their respective weights or coefficients. The overarching ABSA system can be represented as:

$$ABSA_{\text{Bengali}} = \sum_{i=1}^{n} wi \cdot Xi \qquad (1)$$

Where $n$ represents the total number of components in the ABSA system.

In summary, this research's core problem statement revolves around advancing the ABSA in the Bengali language, incorporating transformer models, improving accuracy, and introducing innovative machine learning-deep learning hybrid models. It strives to fill a significant void and enhance sentiment analysis in Bengali, which has implications for applications such as customer feedback analysis and product reviews.

## Methods and materials

This study presents an innovative ABSA methodology, tRF-BERT, which leverages the synergistic strengths of two powerful models: BERT and RF. The research begins by thoroughly curating and preparing publicly accessible datasets encompassing Bengali Cricket and Restaurant reviews. Subsequently, the study diverges into two pivotal tasks: aspect categorization and sentiment evaluation within Bengali text. Aspect categorization is facilitated through the hybrid tRF-BERT model, while sentiment assessment employs the same hybrid model.

A systematic cross-validation strategy encompassing both the RoBERTa and BERT models was adopted to guarantee the stability and dependability of our research findings. This rigorous and iterative validation process enhances the trustworthiness of the empirical results, facilitating a comprehensive exploration of ABSA within the realm of Bengali text. The entirety of the process is illustrated in Fig 1.

### Data collection and preprocessing

Data collection and preprocessing are fundamental steps in machine learning, ensuring that the data are suitable for modelling and high quality. After identifying and selecting the data sources, the data were cleaned to eliminate errors and inconsistencies. Finally the data are split into training and testing sets for model development and evaluation.

**Data source.** The study used publicly available datasets for the first Bengali ABSA model, focusing on Cricket and Restaurant reviews, as it reflects what Bengali people talk about most —their love for their favourite game, cricket and their culinary passion and they often share their sentiments about these topics on social platforms. Analyzing sentiment allows us to understand their views on these everyday topics. Tables 2 and 3 provide an overview of the datasets.

**Data cleaning.** To ensure data quality, a rigorous data cleaning pipeline was employed. This involved cleaning the raw data to eliminate irrelevant information, including non-textual elements such as HTML tags and special characters. Text data were tokenized into sentences, paragraphs, or phrases, depending on the granularity required for ABSA. The dataset was then annotated with aspect and polarity labels to make it suitable for supervised machine-learning tasks.

**Dataset splitting.** To uphold the fidelity of model assessment, the dataset was partitioned into two distinct sets: training and testing. The training set comprised approximately 80% of the data, while the remaining 20% was allocated to the testing set.

### Tokenization and embeddings

1. In the context of the BERT model, the 'bert-base-uncased' tokenizer, sourced from the Transformers library, is employed to partition the text into discrete tokens. These tokens are methodically processed, adhering to predetermined length constraints through truncation or padding techniques. Furthermore, categorical encoding of the aspect and sentiment labels was executed. Feature extraction, while implicitly enacted, transpires during the process of fine-tuning a previously pre-trained BERT [14] model. This pertains to the model's intrinsic capacity to adapt its internal weightings to discern and encapsulate salient textual representations.

2. In the case of RoBERTa, the 'roberta-base' tokenizer is engaged in tokenizing textual content. The tokenized sequences undergo rigorous management, encompassing operations such as truncation, padding, and alignment, all aligned with the prescribed length criteria. Concurrently, aspect and sentiment labels were transformed into a categorical format to

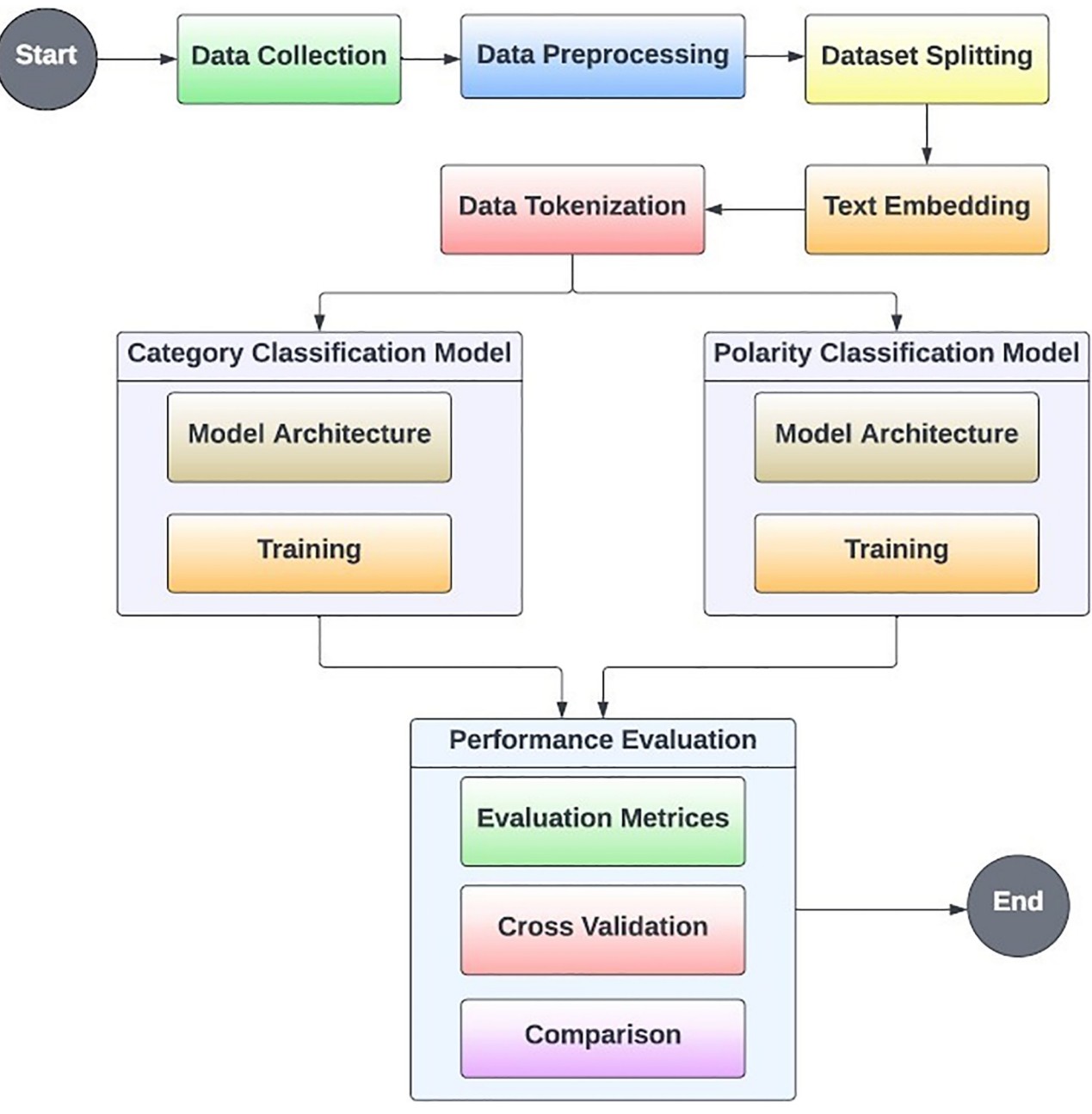

**Fig 1. Aspect-based sentiment analysis process.**

facilitate subsequent analytical procedures. The nucleus of feature extraction becomes manifest during the fine-tuning of a previously pre-trained RoBERTa [24] model. The model self-adjusts its internal weights within this iterative paradigm, which is essential for encoding and discriminating significant textual representations.

3. Within the ambit of the tRF-BERT hybrid model, a process is initiated by encoding labels into a categorical format to expedite the subsequent analytical modeling. Notably, a pretrained BERT [14] model was invoked for the task of category classification. This model is

**Table 2. Statistics of cricket dataset.**

| Aspect Category | Sentiment Polarity | | | Total |
|---|---|---|---|---|
| | Positive | Negative | Neutral | |
| Batting | 137 | 388 | 55 | 580 |
| Bowling | 153 | 144 | 32 | 329 |
| Team | 165 | 501 | 65 | 731 |
| Team Management | 23 | 292 | 14 | 329 |
| Other | 88 | 827 | 95 | 1010 |
| **Total** | | | | 2979 |

**Table 3. Statistics of restaurant dataset.**

| Aspect Category | Sentiment Polarity | | | Total |
|---|---|---|---|---|
| | Positive | Negative | Neutral | |
| Food | 499 | 125 | 86 | 710 |
| Price | 101 | 59 | 15 | 175 |
| Ambience | 137 | 52 | 42 | 231 |
| Service | 185 | 117 | 31 | 333 |
| Miscellaneous | 299 | 119 | 192 | 610 |
| **Total** | | | | 2059 |

subjected to compilation with a legacy optimizer, and its fine-tuning is devoid of dropout layers across a span of ten epochs. Feature extraction is intrinsically intertwined with the fine-tuning procedure of the BERT model, culminating in the model's harmonization of its internal weightings to apprehend distinctive textual attributes during the iterative training phases. Furthermore, textual data undergo vectorization via a Term Frequency-Inverse Document Frequency (TF-IDF) vectorizer, a preparatory step that renders the data amenable to processing by the RF algorithm. The resultant amalgamation of BERT-derived features and machine learning predictions constitutes a composite feature vector, that fulfills the role of input within the final hybrid model.

The utilization of 'bert-base-uncased' and 'roberta-base' tokenizers proved advantageous for our study's small Bengali datasets (Cricket: 2,979, Restaurants: 2,059). Despite being trained in English, these models employ subword tokenization, which extends their functionality to Bengali text. Subword tokenization disassembles words into smaller units that can represent the components of words from diverse languages. This capability enables models to handle unseen Bengali words to some degree, particularly in scenarios with limited data, such as ours. Leveraging these models potentially saves time and effort by obviating the need to develop a custom tokenizer from scratch. Furthermore, they aid in preventing overfitting on small datasets by using regularization techniques. In the case of TF-IDF, the data were also tokenized before further processing, as proper tokenization is required for efficient TF and IDF calculation.

## Aspect and sentiment prediction models

Both the aspect and sentiment classification models were developed by harnessing the capabilities of a hybrid tRF-BERT model. This hybrid model leverages the advantages of both RF and BERT, a cutting-edge NLP framework, to establish a unified method for performing aspect and sentiment classification.

**tRF-BERT model.** The "tRF-BERT" hybrid model combines deep learning and traditional machine learning approaches to advance sentiment classification. It fine-tunes an already pre-trained BERT [14] model, renowned for understanding complex language patterns, for sentiment analysis. Simultaneously, it employs a versatile RF model, optimizing it through TF-IDF vectorization and hyperparameter tuning. Notably, the model integrates predictions from BERT and RF, creating composite feature vectors for the data points. These features were used to construct a neural network model with specific input layers for sentiment classes and RF predictions, thereby enhancing pattern recognition with ReLU activation and dropout regularization. In summary, "tRF-BERT" synergizes the strengths of deep learning and traditional machine learning for to improve sentiment analysis.

In addition to the tRF-BERT model, this ABSA study incorporated two alternative models based on BERT and RoBERTa for cross-validation. The following are concise descriptions of these additional algorithms:

**BERT model.** The BERT model, which is known for its contextual understanding of text, was employed for sentiment analysis. It utilizes a pre-trained neural network architecture to capture bidirectional relationships in the text data [14]. Fine-tuning the BERT model allowed it to adapt to specific ABSA tasks. BERT's attention mechanism and multi-layer representation make extracting aspects and their associated sentiments effective. An overview of the BERT model is presented in the following sections.

*Setup*. As an input, BERT takes two sequences of tokens. Assume X = {$x_1$, ..., $x_n$} and Y = {$y_1$, ..., $y_n$} are two segments. Here M + N < T controls the maximum sequence length during the training. Here, M refers to the maximum number of tokens allowed in the first segment (X) and N refers to the maximum number of tokens allowed in the second segment (Y) of the input sequence. T defines the absolute maximum length for the entire combined input sequence, including special tokens such as [CLS], [SEP], and [EOS]. Unique tokens are used to separate the two segments X and Y, which are supplied to the BERT as one input sequence [CLS], *X*, [SEP], *Y*, [EOS]. [24] Here, the [CLS] or classifier token marks the beginning of the input sequence and is used for the classification tasks. This helps the model to understand how the entire sequence relates to the task at hand. The [SEP] or separator token separates two segments (X and Y) within the input sequence. It describes the model where one segment ends and the other begins. The [EOS] or the End of Sentence is an optional token. This explicitly signals to the model the end of the entire input sequence. Although some models inherently understand this from the padding tokens used to reach the maximum sequence length.

The BERT employs a widely used transformer design [33]. Assume that Q, K, and V are the three input parameters where $Q \in R^{n \times d_k}$ represents queries, $K \in R^{m \times d_k}$ represents keys and $V \in R^{m \times d_v}$ represents values. The matrix output is computed as follows:

$$\text{Attention}(Q, K, V) = \text{softmax}\left(\frac{QK^T}{\sqrt{d_k}}\right) V \tag{2}$$

In attention mechanisms, query (Q), key (K), and value (V) vectors are essential abstractions for calculating and reasoning attention. Q or queries represent informational needs derived from a specific word or phrase processed at a particular position in the sequence (either X or Y). In simpler terms, Q captures what the model needs to know at a specific point to understand the context. K or the keys represent the information available in each part of the input sequence. Keys act as labels or summaries for each element in the sequence, allowing the model to identify relevant information based on the query (Q). V, or values, represent the actual information associated with each element in the sequence. They hold the content itself, which is used to fulfill the informational needs represented by queries (Q). BERT utilizes a

multi-headed attention mechanism, where multiple sets of Q, K, and V parameters are created and processed independently. Both segments (X and Y) are first converted into word embeddings. These embeddings are numerical representations that capture the meaning of each word. Additionally, positional encodings are added to the word embeddings. These encodings help the model to understand the relative position of each word within the sequence (X or Y), which is crucial for tasks such as sentence understanding. These processed word embeddings with positional encodings are then used to generate multiple sets of Q, K, and V for each word in the sequence. This is achieved through linear transformations of the embeddings. Essentially, the Q, K, and V parameters leverage the information from word embeddings and positional encodings within segments X and Y to perform attention and focus on the most relevant parts of the sequence for a specific task. This allows BERT to understand the relationships between words within and across segments, leading to a more comprehensive contextual representation for tasks such as question answering or sentiment analysis.

*Encoder*. Built upon transformer encoders, BERT employs a series of N-stacked encoder blocks. Each block is responsible for discerning the relationships among input representations and transforming them into outputs, which are subsequently forwarded to the next encoder block. These layers comprise two sub-layers: a feed-forward network and a multi-head attention layer. The multi-head attention mechanism enables the model to collectively focus on information across various representation subspaces and positions. However, using only one attention head hampers this collaborative aspect, because averaging diminishes its effectiveness.

$$MultiHead(Q, K, V) = Concat(head_1, \ldots, head_h)W^O$$

where,

$$head_i = Attention(QW_i^Q, KW_i^K, VW_i^V) \tag{3}$$

In addition to the attention sub-layers, each layer in the encoder contains a fully connected feed-forward network, which is applied to each position separately and identically. This consists of two linear transformations with ReLU activation between them.

$$FFN(x) = max(0, xW_1 + b_1)W_2 + b_2 \tag{4}$$

Here, x is the input vector to the FFN layer, $W_1$ is the weight matrix of the first layer, $b_1$ is the bias vector of the first layer, $W_2$ is the weight matrix of the second layer, $b_2$ is the bias vector of the second layer.

Although the linear transformations are the same across different positions, they use different parameters from layer to layer. Another way of describing this is through two convolutions with a kernel size of 1.

Fig 2 illustrates the architecture of a single encoder layer and its two sublayers.

*Training objectives*. BERT used Masked Language Modeling (MLM) and Next Sentence Prediction (NSP) for pre-training. The combined application of MLM and NSP reduces the combined loss function of the two techniques [34].

1. **MLM**: In this step, BERT uniformly selected 15% of the input tokens and replaced the 80% of the selected tokens with [MASK]. From the remaining 20%, a vocabulary token chosen at random replaces 10% of the remaining tokens, leaving the remaining 10% unaltered [34]. The complete sequence is encoded using the BERT attention-based encoder, which predicts masked words only using the context of the other non-masked words.

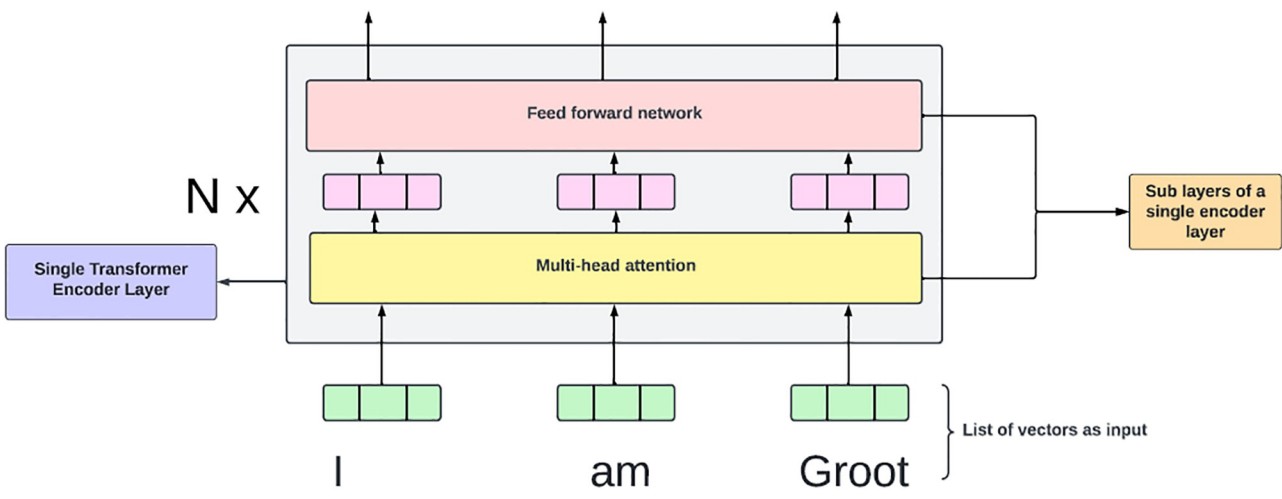

**Fig 2. BERT Encoder architecture.**

2. **NSP**: NSP binary classification loss predicts whether the second sentence follows the first sentence in the original text [34]. The model assigns a label of"IsNext" if the second sentence follows the first sentence in the original text, and"NotNext" if the second sentence does not follow the first one in the original text. Each label has an equal probability of occurence.

*Optimization*. The Adam optimizer [35] was used to optimize BERT with certain parameters set, such as $\beta_1 = 0.9$, $\beta_2 = 0.999$, $\epsilon = $ 1e-6, L2 weight decay of 0.01 and a learning rate schedule of 1e-4 over 10,000 steps, followed by linear decay [24]. During training, it uses a dropout of 0.1 on all layers, attention weights, and the Gaussian Error Linear Unit (GELU) activation function [36]. A significant number of updates and 256 sequences per minibatch with a maximum length of 512 tokens are required for the effective pre-training of the models [24].

*Data*. A vast dataset consisting of 2.5 billion words from English Wikipedia and 800 million words from BOOKCORPUS [37] was used to train BERT. With the help of this vast amount of training data, BERT can acquire thorough linguistic representations and patterns, which helps it accomplish a variety of NLP tasks remarkably accurately.

**RoBERTa model.** RoBERTa (A Robustly Optimized BERT Pretraining Approach) is another variant of the BERT model. RoBERTa builds upon the BERT architecture using further optimization techniques [24]. It enhances performance by utilizing larger mini-batches, training data, and sequences. RoBERTa excels in understanding the nuances of text data and is particularly effective for sentiment analysis. It is known for its robustness and improved results, particularly in cases with limited labeled data.

**RF.** The RF algorithm is widely employed in machine learning to effectively address the regression and classification challenges. Leo Breiman first developed this algorithm [38]. RF produces results by combining the outputs of multiple tree-structure classifiers. In 2001, Leo Breiman defined RF as [38]:

"A RF is a classifier consisting of a collection of tree-structured classifiers {h(x, $\Theta_k$), k = 1, ...} where the {$\Theta_k$} are independent identically distributed random vectors, and each tree casts a unit vote for the most popular class at input x."

The final decision function of RF is [39]:

$$H(x) = \arg\max \Sigma_{i=1}^{k} I(h_i(x) = Y) \tag{5}$$

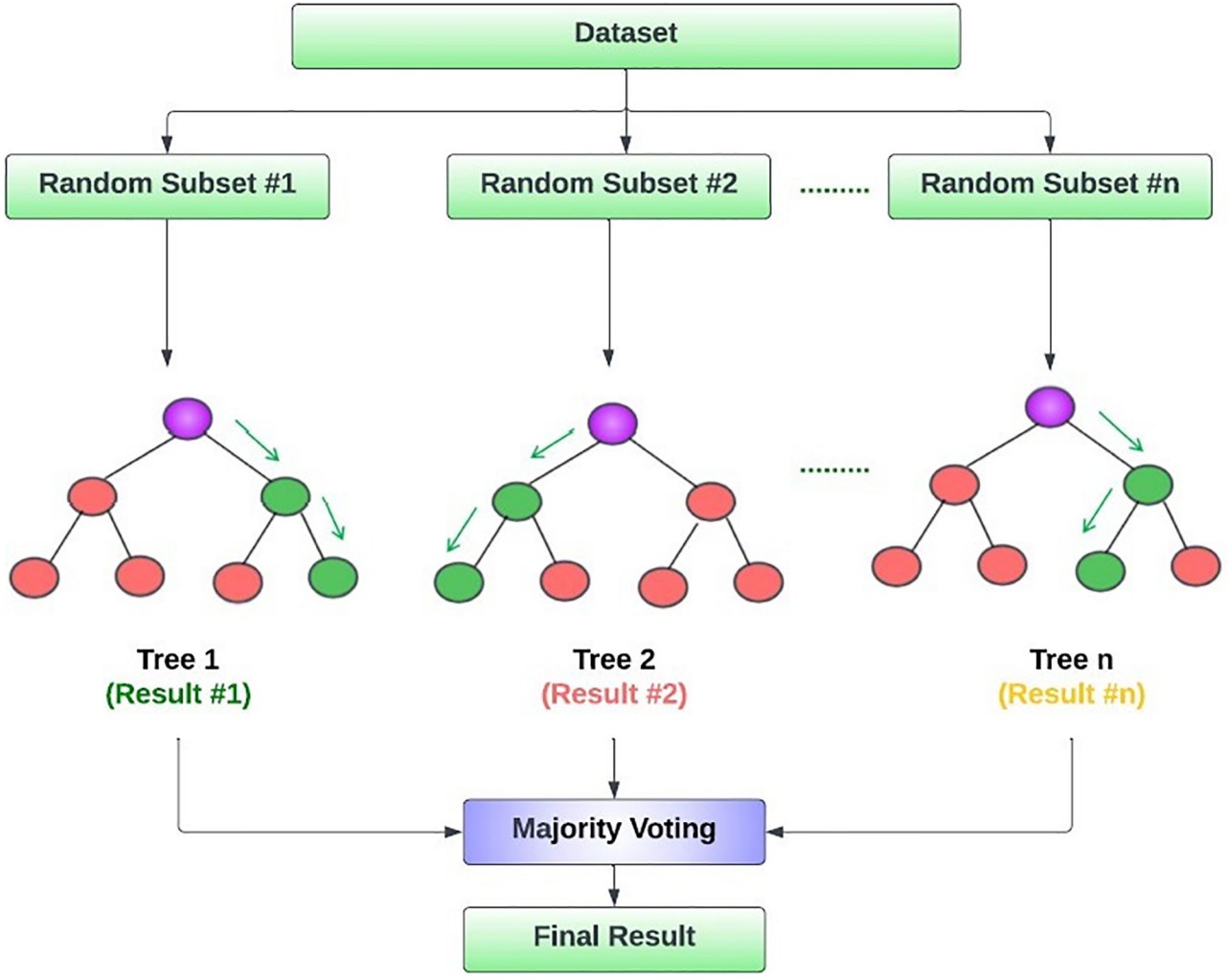

**Fig 3. Random forest architecture.**

Here, $H(x)$ is the classification model of the classifier, $h_i$ represents a single decision tree, $Y$ represents the classification label (output variable), and $I(*)$ represents the indicator function.

A random forest constructs an ensemble of decision trees, with each tree being trained on a randomly selected subset of the dataset. Fig 3 illustrates the fundamental concept underlying the operational procedure of the Random Forest algorithm.

The BERT and RoBERTa models are recognized for their state-of-the-art performance in natural language understanding tasks, including sentiment analysis. Their use alongside the tRF-BERT model showcases a comprehensive approach to ABSA that combines the strengths of various deep learning architectures. Cross-validation ensures the robustness and generalization of the sentiment analysis results.

## Hyperparameters

Tables 4–6 list the values of the different essential hyperparameters that were finalized for use in the proposed models. These hyperparameters were randomly chosen for the first trial; however, their values were adjusted iteratively until an optimal value was achieved. For example,

**Table 4. Hyperparameters for tRF-BERT.**

| Hyperparameter | Value |
| --- | --- |
| Learning Rate (Adam Optimizer) | 2e-5 |
| Batch Size | 32 |
| Early Stopping Patience | 3 |
| Learning Rate Reduction Factor | 0.2 |
| Learning Rate Reduction Patience | 2 |
| Number of Estimators (RF) | 100 |
| Maximum Features for TF-IDF | 1000 |
| Activation Function (Dense Layers) | relu |
| Loss Function | Sparse_categorical_crossentropy |

**Table 5. Hyperparameters for BERT.**

| Hyperparameter | Value |
| --- | --- |
| Learning Rate (Adam Optimizer) | 2e-5 |
| Batch Size | 32 |
| Early Stopping Patience | 3 |
| Learning Rate Reduction Factor | 0.2 |
| Learning Rate Reduction Patience | 2 |
| Activation Function (Dense Layers) | relu |
| Loss Function | sparse_categorical_crossentropy |

**Table 6. Hyperparameters for RoBERTa.**

| Hyperparameter | Value |
| --- | --- |
| Learning Rate (Adam Optimizer) | 2e-5 |
| Batch Size | 32 |
| Early Stopping Patience | 3 |
| Learning Rate Reduction Factor | 0.2 |
| Learning Rate Reduction Patience | 2 |
| Activation Function (Dense Layers) | relu |
| Loss Function | sparse_categorical_crossentropy |

the learning rate of the experiment started with a value of 1e-3, but after examining the accuracy and error rate, it was adjusted linearly, from 1e-3 to 1e-5, until optimal performance was achieved. Regardless of how much one hyperparameter is changed, its effect on the model's performance is limited to some extent. The combination of all hyperparameters is crucial. Similarly, other hyperparameters were tested; for instance, the initial batch size was 16, but as the batch size increased, so did the performance. Therefore, it was adjusted and set accordingly. In the hyperparameter tuning process of random forest, parameters such as the number of trees (n estimators) and the maximum depth were determined through iterative trial and error. For this study, the values of n estimators = 100 and max depth = 10 were selected after multiple trials.

The hyperparameters used for tuning tRF-BERT, BERT, and RoBERTa models are listed in Tables 4–6:

## Proposed tRF-BERT model

- **BERT-based model**: The first component of the tRF-BERT hybrid model involves fine-tuning pre-trained BERT model for sequence classification. The BERT model captures the text data's intricate language patterns and contextual information. It was trained on a labeled dataset for sentiment classification. The output layer of the BERT model corresponds to the number of unique sentiment classes in the dataset, facilitating multi-class prediction.

- **RF model**: In parallel, an RF model is introduced to complement the BERT-based approach. Text data are vectorized using Term Frequency-Inverse Document Frequency (TF-IDF) to generate numerical features. The RF model is subjected to hyperparameter tuning through a grid search to optimize its performance [40].

- **Final tRF-BERT model**: This methodology's key innovation lies in combining predictions from both the BERT-based and RF models to create a feature vector for each data point. The hybrid model was constructed as a neural network with an input layer that accounted for the number of classes and additional input for RF prediction. A hidden layer, employing ReLU activation and dropout regularization, enhances the model's capability to extract complex patterns in the data. The output layer employs softmax activation for multi-class prediction.

The Random Forest model was utilized alongside the BERT model to incorporate different perspectives and capture a wider range of features from the text data. Despite the known limitations of Random Forest models in ABSA tasks, such as potentially lower performance compared to deep learning models such as BERT, its inclusion allows for a diverse ensemble approach. The BERT model, a deep learning architecture pre-trained on large corpora, excels in capturing complex linguistic patterns and contextual information, making it well-suited for sentiment analysis tasks. Its ability to understand the semantics and context of text contributes to accurate sentiment predictions.

On the other hand, the Random Forest model, although simpler in structure, offers strengths in handling non-linear relationships and capturing interactions among features. It can effectively identify important features and provide robust predictions, particularly in scenarios with limited data or when dealing with noisy datasets. By integrating predictions from both models, the hybrid approach harnesses the deep learning capabilities of BERT while also benefiting from the robustness and interpretability of the Random Forest model. While weighting the predictions based on performance could further optimize the hybrid model, equal importance is given to maintain simplicity and avoid introducing additional complexity. Given the diversity in the strengths of the two models and their potential to capture different aspects of sentiment, the equal weighting approach aims to strike a balance and ensure that the hybrid model benefits from both perspectives without bias. Implementing a weighting scheme in which the predictions from each model are assigned weights based on their respective performance metrics is feasible. We plan to incorporate this approach in future work.

The architecture of this tRF-BERT model is depicted in Fig 4.

## Training

The hybrid models were fine-tuned and trained for improved performance:

**Optimizer.** The Adam optimizer was employed for gradient-based updates.

**Loss function.** Sparse categorical cross-entropy loss function facilitated training.

**Increased epochs.** Training was extended to 20 epochs to ensure model convergence.

**Larger batch size.** A batch size of 64 was selected for efficient training.

**Validation split.** A 20% validation split was used for model evaluation.

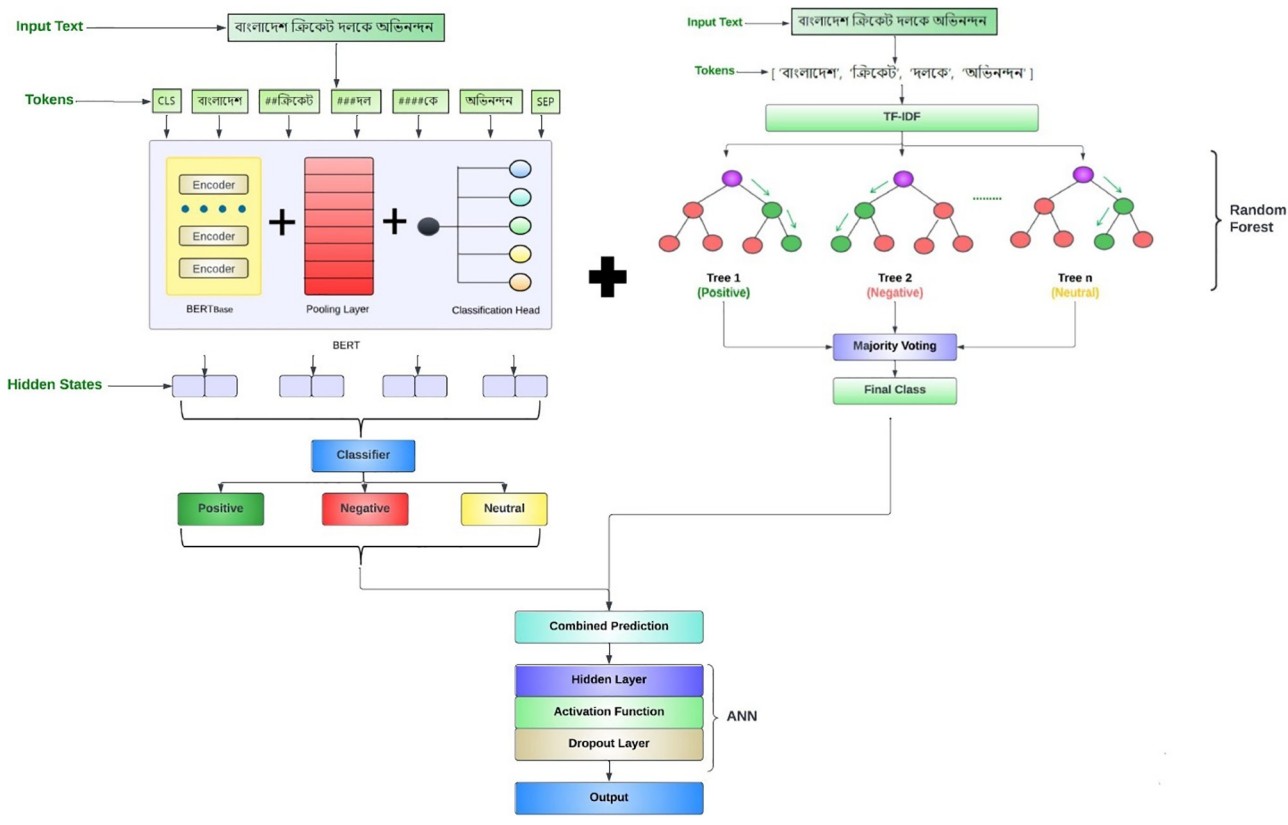

**Fig 4. tRF-BERT Architecture.**

### Performance evaluation

**Evaluation metrics.**   The performance of each model, including the category and polarity classification, was rigorously evaluated using standard metrics. These metrics included accuracy, precision, recall, F1-score, and comprehensive classification reports. The evaluation framework provides an intricate understanding of a model's competence in precise categorization and sentiment polarity determination.

**Cross-validation.**   A systematic cross-validation strategy encompassing both BERT and RoBERTa models was adopted to ensure the robustness and reliability of the research findings. This approach entails partitioning the dataset into multiple folds and iteratively training and evaluating models. A value of k = 5 was used for the cross-validation. The cross-validation process enhances the trustworthiness of the empirical results, reduces the risk of overfitting, and ensures the generalization of model performance.

**Comparison.**   An extensive comparative analysis was conducted to discern the performance nuances of the individual BERT-based and hybrid models. This empirical examination is pivotal in determining which model variant excels in ABSA tasks, setting the stage for comprehensive ABSA within the Bengali text.

### Result analysis

This section discusses the findings of the proposed hybrid tRF-BERT model and the results obtained from the BERT and RoBERTa models. We also present a comparative analysis of the findings of previous studies.

### Preparing environment

Our research necessitated a robust computing infrastructure for deep learning experiments, consisting of an Intel Core i7 processor and 16 GB of RAM. We implemented our code using Python and the KERAS library, with TensorFlow serving as the underlying computational framework. This configuration facilitated our engagement in intricate deep-learning tasks and the subsequent analysis of complex model structures.

## Results

We thoroughly scrutinized the outcomes for both subtasks using established evaluation metrics that are crucial for evaluating the classification task performance. These metrics are foundational for understanding the efficacy of our models in accurately classifying instances. Before discussing the metrics, it is essential to elucidate the critical components that they rely upon.

- True Positives (TP): Correctly identified instances of a specific class [41]

- True Negatives (TN): Instances correctly recognized as not belonging to the class [41].

- False Positives (FP): Instances mistakenly identified as belonging to the class [41].

- False Negatives (FN): Instances incorrectly classified as not belonging to the class [41].

With these fundamental components elucidated, we can now explore the core evaluation metrics that are instrumental to our study:

1. Accuracy: Accuracy offers a high-level perspective of model performance, indicating the ratio of correctly predicted instances (encompassing both TP and TN) to the entire dataset [41].

$$Accuracy = \frac{TP + TN}{TP + FP + TN + FN} \qquad (6)$$

2. Precision: This quantifies the model's ability to accurately identify instances of a particular class without erroneously classifying others. Calculated as TP divided by the sum of TP and FP, high precision values indicate a reduced rate of false positives [42].

$$Precision = \frac{TP}{TP + FP} \qquad (7)$$

3. Recall: Recall, also known as sensitivity or the true positive rate, measures the model's proficiency in correctly identifying instances of a specific class. It is computed as TP divided by the sum of TP and FN, and high recall values signify a diminished incidence of false negatives [42].

$$Recall = \frac{TP}{TP + FN} \qquad (8)$$

4. F1-Score: The F1-Score is a pivotal metric that serves as a balanced measure of the precision and recall. This metric assumes particular importance when dealing with datasets characterized by class imbalances. Computed as the harmonic mean of precision (TP / [TP + FP])

and recall (TP / [TP + FN]), the F1-Score provides a robust and holistic measure of the overall performance of our classification models [41].

$$F1\text{-}score = \frac{2 * (Recall * Precision)}{Recall + Precision} \tag{9}$$

Tables 7 and 8 show the aspect and sentiment classification results for the Cricket dataset, respectively. In the aspect classification task, tRF-BERT demonstrated the highest accuracy of 0.89, closely followed by RoBERTa at 0.88, with BERT trailing slightly at 0.83. RF's accuracy is 0.65. A similar trend was observed for precision, recall, and F1-score, where tRF-BERT consistently outperformed the other models.

These results suggest that tRF-BERT exhibits superior capabilities in accurately classifying aspects within the text, with strong precision and recall trade-offs, resulting in a commendable F1-score of 0.85. In contrast, BERT and RoBERTa, while still achieving reasonable performance, lagged slightly behind in terms of accuracy and F1-score. In the sentiment classification task, tRF-BERT again emerged as the frontrunner, with an accuracy of 0.93, surpassing BERT, RoBERTa and RF, achieving an accuracy of 0.86. tRF-BERT maintains this lead in terms of precision, recall, and F1-score, further indicating its robustness in sentiment classification. The results underline the superiority of tRF-BERT in both aspect and sentiment classification, suggesting its potential for enhancing the understanding and analysis of textual data. BERT and RoBERTa also exhibit strong performance, making them viable alternatives in various NLP tasks.

Tables 9 and 10 show the aspect and sentiment classification results for the Restaurant Dataset. In the aspect classification task, tRF-BERT emerged as the leader with the highest

**Table 7. Aspect detection for cricket dataset.**

| Algorithm | Accuracy | Precision | Recall | F1-score |
|---|---|---|---|---|
| tRF-BERT | **0.89** | **0.83** | **0.89** | **0.85** |
| RoBERTa | 0.88 | 0.80 | 0.88 | 0.84 |
| BERT | 0.83 | 0.76 | 0.83 | 0.78 |
| RF | 0.65 | 0.48 | 0.65 | 0.55 |

**Table 8. Sentiment classification for cricket dataset.**

| Algorithm | Accuracy | Precision | Recall | F1-score |
|---|---|---|---|---|
| tRF-BERT | **0.93** | **0.86** | **0.93** | **0.90** |
| RoBERTa | 0.86 | 0.73 | 0.86 | 0.79 |
| BERT | 0.86 | 0.73 | 0.86 | 0.79 |
| RF | 0.66 | 0.67 | 0.66 | 0.66 |

**Table 9. Aspect detection for restaurant dataset.**

| Algorithm | Accuracy | Precision | Recall | F1-score |
|---|---|---|---|---|
| tRF-BERT | **0.92** | **0.87** | **0.92** | **0.89** |
| RoBERTa | 0.91 | 0.84 | 0.91 | 0.87 |
| BERT | 0.87 | 0.78 | 0.87 | 0.82 |
| RF | 0.60 | 0.47 | 0.60 | 0.62 |

**Table 10. Sentiment classification for restaurant dataset.**

| Algorithm | Accuracy | Precision | Recall | F1-score |
|---|---|---|---|---|
| tRF-BERT | **0.95** | **0.91** | **0.95** | **0.93** |
| RoBERTa | **0.95** | **0.91** | **0.95** | **0.93** |
| BERT | 0.94 | 0.88 | 0.93 | 0.91 |
| RF | 0.60 | 0.45 | 0.60 | 0.52 |

accuracy of 0.92, closely followed by RoBERTa at 0.91, whereas BERT slightly lagged behind at 0.87. RF performed poorly compared with the other three methods, with an accuracy of 0.60. A similar pattern was evident in precision, recall, and F1-score, with tRF-BERT consistently outperforming the other models. These results underscore the superior capabilities of tRF-BERT in accurately classifying aspects within the text, with strong precision and recall trade-offs that yield a commendable F1-score of 0.89.

While Accuracy is sufficient for balanced datasets, imbalanced datasets require a more nuanced approach. In such cases, Precision, Recall, and F1-Score provide valuable insights. By considering these metrics, we gain a comprehensive understanding of how well our model identifies every case, even when the classes are not evenly distributed. Although there was some imbalance in the dataset, we were able to achieve the expected results without requiring the application of methods such as oversampling, undersampling, and class weighting. However, we plan to incorporate these methods in further experimentation.

In contrast, BERT and RoBERTa, while still delivering reasonable performance, slightly lag in terms of accuracy and F1-score. However, in the sentiment classification task, tRF-BERT and RoBERTa took the lead with the same accuracy score of 0.95. A similar trend was observed for precision, recall, and F1-score, with tRF-BERT and RoBERTa consistently outperforming BERT and RF. These results indicate that tRF-BERT and RoBERTa exhibit stronger capabilities in accurately classifying sentiment within the text, with higher precision and recall trade-offs, resulting in superior F1-scores of 0.93. In this context, while still performing reasonably well, BERT slightly lags behind in terms of F1 score and accuracy.

Table 11 lists the model runtimes. For both the Cricket and Restaurant datasets, the runtimes for the RoBERTa and BERT models were comparable, with slight variations in execution times across tasks. The Hybrid model consistently showed similar or slightly longer runtimes than RoBERTa and BERT. The RF had the lowest runtime. Considering the enhancement in performance that the hybrid model offers compared with RoBERTa and BERT, we find the slightly longer execution time to be justifiable. In situations where maximizing performance is paramount, prioritizing accuracy over execution time becomes essential. Therefore, a trade-off between runtime and performance gain is warranted, because the primary objective is to achieve superior results for the given task. Hence, opting for the hybrid model, despite the extended runtime, can be justified given its performance improvements. While experimenting with approaches like BERT Tokenizer and Word2Vec, we found that for our specific datasets,

**Table 11. Runtime comparison.**

| Task | RoBERTa | BERT | RF | tRF-BERT |
|---|---|---|---|---|
| Cricket (Aspect detection) | 3m 18s | 3m 1s | 2m | 3m 8s |
| Cricket (Sentiment classification) | 3m 54s | 2m 58s | 1m 58s | 3m 1s |
| Restaurent (Aspect detection) | 2m 58s | 3m | 1m | 3m |
| Restaurent (Sentiment classification) | 2m 57s | 2m 15s | 1m | 2m 2s |

**Table 12. Accuracy comparison with different tokenizer.**

| Task | BERT Tokenizer | Word2vec | TF-IDF |
|---|---|---|---|
| Cricket (Aspect detection) | 0.72 | 0.63 | 0.89 |
| Cricket (Sentiment classification) | 0.77 | 0.69 | 0.93 |
| Restaurent (Aspect detection) | 0.79 | 0.73 | 0.92 |
| Restaurent (Sentiment classification) | 0.81 | 0.78 | 0.95 |

**Table 13. Accuracy of different models while experimenting for building hybrid model.**

| Task | SVM-BERT | LR-BERT | tRF-BERT |
|---|---|---|---|
| Cricket (Aspect detection) | 0.72 | 0.63 | 0.89 |
| Cricket (Sentiment classification) | 0.77 | 0.69 | .93 |
| Restaurent (Aspect detection) | 0.70 | 0.73 | 0.92 |
| Restaurent (Sentiment classification) | 0.81 | 0.78 | 0.95 |

TF-IDF yielded better results in our hybrid model. The accuracy of each approach for tRF-BERT is given in Table 12.

The possible reasons are:

- Smaller Datasets: TF-IDF can be particularly effective with smaller datasets, such as ours, where capturing the statistical relationships between words might be more beneficial than complex language models.

- Domain-Specific Focus: Our datasets have a strong domain focus, and TF-IDF can be used to identify relevant keywords and their importance within that domain, potentially leading to good performance.

To address scalability concerns, careful consideration of data preprocessing and memory management techniques is essential.

We experimented with other models such as Logistic Regression (LR), and SVM with BERT to build other hybrid models. We found that the combination of BERT and RF worked better than the others. The experimental results are listed in Table 13.

## Comparison with previous works

To the best of our knowledge, previous studies have predominantly focused on subtask 1, which involves aspect detection, for both cricket and restaurant datasets. Here, we offer a comparative assessment of our research concerning subtask 1 for both datasets compared to prior research.

1. **1. Cricket dataset**: Compared with previous works by Rahman *et al*., Haque *et al*., and F. A. Naim, our research represents substantial progress in the case of detecting aspects of the cricket dataset. Employing advanced models such as BERT, RoBERTa, and tRF-BERT, our research attained notably higher accuracy rates (0.83, 0.88, and 0.89, respectively) and robust F1-scores (0.78, 0.84, and 0.85). These results underscore the efficacy of these models in accurately identifying the cricket-related aspects. On the other hand, the earlier studies conducted by Rahman *et al*. and Haque *et al*. mostly used traditional machine learning models, resulting in relatively lower F1-scores, highlighting the potential limitations of these traditional methods. Notably, F. A. Naim's research achieved competitive F1-scores

by employing SVM and CNN. Our findings, rooted in deep learning techniques, emphasize substantial progress in aspect detection capabilities, enabling more precise and context-aware analysis of cricket-related textual data.

2. **2. Restaurant dataset**: Within the restaurant dataset, our research emerges as a notable frontrunner when compared to prior studies conducted by Rahman *et al.*, Haque *et al.*, and F. A. Naim. Leveraging models such as BERT, RoBERTa, and tRF-BERT, our research achieves significantly higher accuracy levels (0.87, 0.91, and 0.92, respectively) and robust F1-scores (0.82, 0.87, and 0.89). These results highlight the proficiency of these models in accurately identifying restaurant-related aspects. In contrast, earlier research by Rahman *et al.* and Haque *et al.* predominantly relied on traditional machine learning models, yielding comparatively lower F1-scores, indicating constraints in their ability to capture nuanced aspects. While F. A. Naim's research achieved competitive F1-scores with the use of SVM and CNN, our findings underscore substantial advancements in aspect detection capabilities, offering improved precision and context-aware analysis of textual data, particularly within the restaurant domain. Table 14 compares the results of our research with those of previous studies.

**Table 14. Evaluating this study in relation to prior research on aspect detection in Bengali ABSA.**

| Research | Dataset | Algorithm | Accuracy | F1-score |
|---|---|---|---|---|
| Rahman *et al.* [16, 18] | Cricket | RF | 0.25 | 0.37 |
| | | SVM | 0.19 | 0.35 |
| | | CNN | 0.81 | 0.51 |
| Rahman *et al.* [16, 18] | Restaurant | RF | 0.30 | 0.33 |
| | | SVM | 0.29 | 0.38 |
| | | CNN | 0.83 | 0.64 |
| Haque *et al.* [28] | Cricket | RF | - | 0.37 |
| | | SVM | - | 0.35 |
| | | LR | - | 0.34 |
| Haque *et al.* [28] | Restaurant | RF | - | 0.35 |
| | | SVM | - | 0.39 |
| | | LR | - | 0.43 |
| F. A. Naim [29] | Cricket | SVM | - | 0.48 |
| | | CNN | - | 0.59 |
| | | RF | - | 0.41 |
| F. A. Naim [29] | Restaurant | RF | - | 0.35 |
| | | SVM | - | 0.52 |
| | | CNN | - | 0.67 |
| This research | Cricket | BERT | 0.83 | 0.78 |
| | | RoBERTa | 0.88 | 0.79 |
| | | tRF-BERT | 0.89 | 0.85 |
| | | RF | 0.65 | 0.55 |
| This research | Restaurant | BERT | 0.87 | 0.82 |
| | | RoBERTa | 0.91 | 0.87 |
| | | tRF-BERT | 0.92 | 0.89 |
| | | RF | 0.60 | 0.62 |

## Discussion

The proposed tRF-BERT model outperformed both the BERT and RoBERTa models for aspect detection and sentiment classification. A comprehensive explication delineating the enhanced performance of the hybrid tRF-BERT model is subsequently provided.

- The key innovation of the tRF-BERT model lies in combining the predictions of the BERT model and the RF models. This model captures the complex patterns in the data through the BERT-based component and the feature engineering capabilities of the RF component. By combining the best features of both models, this hybrid technique enables the model to produce more reliable and accurate predictions.

- The ability of BERT to capture complex language patterns and contextual information in text data plays a crucial role in effectively identifying aspects and sentiments. The BERT model was pre-trained on a massive dataset, allowing it to learn these patterns and make more informed predictions. By combining BERT with an RF model that uses TF-IDF for text representation, the tRF-BERT model leverages the strengths of both approaches.

- Our tRF-BERT model uses RF, which is an ensemble learning method. One of its best features is that it can reduce overfitting and increase generalization by combining the predictions from multiple decision trees. This is particularly helpful when working with noisy or high-dimensional datasets. Thus it can be used to identify patterns that BERT might overlook, making it a useful complement to the tRF-BERT model.

- The tRF-BERT model learns to dynamically optimize the relevance of each component by incorporating predictions from both the RF and BERT-based model into a neural network for final predictions. The use of ReLU activation and dropout regularization in the hidden layer of the neural network can enhance the ability of the model to recognize complex patterns between aspects and sentiments and prevent overfitting.

- Finally, the tRF-BERT model employs softmax activation in the output layer, which is specifically designed for multi-class classification. This activation function ensures that the model generates probability distributions over potential sentiment classes to guarantee that the model can successfully assign probabilities to each sentiment class. This is important for an accurate sentiment classification.

In summary, the tRF-BERT model performed better than BERT and RoBERTa because of its hybrid methodology, which combines the strengths of the RF and BERT models. This hybrid approach, coupled with a deep learning architecture and softmax activation, allows the tRF-BERT model to achieve enhanced accuracy, robustness, and interpretability for aspect detection and sentiment classification tasks.

## Conclusions

In conclusion, the tRF-BERT methodology presented in this study has substantial implications for Bengali ABSA research and the wider domain of the Bangla NLP. This innovative approach propels ABSA studies in Bengali to new heights and marks a pivotal milestone in comprehending and dissecting sentiment in Bengali text. Notably, the exceptional performance of tRF-BERT, as reflected in its high accuracy and F1-scores, underscores its potential as a potent tool for meticulous and context-aware sentiment analysis in Bengali. The methodological rigor of the method, bolstered by systematic cross-validation, significantly reinforces the credibility of its findings, establishing a robust foundation for forthcoming ABSA research in Bengali. Importantly, tRF-BERT's superior performance, surpassing that of other ABSA models on the

same dataset, highlights its pioneering role in pushing the boundaries of sentiment analysis in the Bengali language and its potential to shape the future landscape of ABSA research in this linguistic domain. Furthermore, this methodology serves as a benchmark, providing valuable perspectives for broader Bangla NLP applications and setting the stage for advanced model utilization in various linguistic and domain-specific contexts.

## Author Contributions

**Conceptualization:** Moythry Manir Samia, M. F. Mridha.

**Formal analysis:** Moythry Manir Samia.

**Investigation:** Maksuda Haider Sayma.

**Methodology:** Maksuda Haider Sayma.

**Supervision:** M. F. Mridha.

**Validation:** Md. Mohsin Kabir.

**Visualization:** Md. Mohsin Kabir.

**Writing – original draft:** Shihab Ahmed.

**Writing – review & editing:** M. F. Mridha.

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
