## [Decision Letter · Decision Letter 0]

18 Mar 2024

PONE-D-24-01459tRF-BERT: A transformative approach to aspect-based sentiment analysis in the bengali languagePLOS ONE

Dear Dr. Mridha,

Thank you for submitting your manuscript to PLOS ONE. After careful consideration, we feel that it has merit but does not fully meet PLOS ONE’s publication criteria as it currently stands. Therefore, we invite you to submit a revised version of the manuscript that addresses the points raised during the review process.

We look forward to receiving your revised manuscript.

Kind regards,

Qin Xiang Ng, MBBS, MPH

Academic Editor

PLOS ONE

Journal Requirements:

Additional Editor Comments:

Apologies for the delay in securing reviewers for this manuscript. After reviewing the manuscript as well as the reviewers' comments and feedback, it is quite apparent that major revisions are necessary before the resubmitted manuscript can be considered. There is no guarantee of acceptance. 

1. Given the journal's biomedical and public health focus, some applications of sentiment analysis in public health research should be highlighted in the Introduction section (see: https://pubmed.ncbi.nlm.nih.gov/37376407;
https://pubmed.ncbi.nlm.nih.gov/37358808).

2. Although the authors provided a description of the proposed hybrid transformative Random Forest and Bidirectional Encoder Representations from Transformers (tRF-BERT) model, there is limited explanation on the technical underpinnings, such as the specific architecture details, the interaction between Random Forest and BERT components, and how exactly the hybrid model outperforms its constituent parts.

3. While the manuscript mentions using two open-source benchmark datasets, Cricket and Restaurant, for Aspect-Based Sentiment Analysis, I am unable to find any citations for these datasets, and the authors do not provide further statistics about these datasets (e.g., number of samples, distribution of classes). A more thorough dataset description is necessary.

4. The comparison against existing works primarily focuses on the final performance metrics. A comparison which discusses the nature of the datasets used in those works, model complexities, and computational resources required, would offer a clearer picture of the proposed model's advantages and limitations.

5. The author strongly focuses on F1 score and accuracy for evaluating model performance. Incorporating additional metrics such as Precision-Recall AUC, Matthews Correlation Coefficient, or analysis on the model's performance across different aspects/categories could provide a more comprehensive evaluation.

6. While some hyperparameters are listed, the process of selecting these values or any optimization strategy employed is not discussed. Detailing the hyperparameter tuning process, including the range of values explored, would strengthen the methodological rigor.

Reviewers' comments:

Reviewer's Responses to Questions

**Comments to the Author**

1. Is the manuscript technically sound, and do the data support the conclusions?

Reviewer #1: Yes

Reviewer #2: No

2. Has the statistical analysis been performed appropriately and rigorously? 

Reviewer #1: Yes

Reviewer #2: No

3. Have the authors made all data underlying the findings in their manuscript fully available?

Reviewer #1: Yes

Reviewer #2: No

4. Is the manuscript presented in an intelligible fashion and written in standard English?

Reviewer #1: Yes

Reviewer #2: Yes

5. Review Comments to the Author

Reviewer #1: - The paper provides a technically sound piece of scientific research with data that supports the conclusions.

- The authors present appropriately the statistical analysis.

- The authors provide data underlying the findings in their manuscript.

- This paper is well-written in English with some needed revisions.

Specific comments:

- In the abstract section, there is a sentence “it was clear that all the models used for our work achieved better results than any of the previous work”, what does the “it” word refer to? If it is not unclear, please revise!

- In the sentence “A crucial part of this research involved finding or creating a dataset specifically designed for Bengali aspect-level sentiment analysis” on page 2, the authors mention that this paper creating a dataset. However, the page 7 mentioned that this study used publicly available datasets. Please clarify this issue!

- There is the sentence “it’s crucial to consider the semantics of the given aspect as a new and distinct piece of information, separate from the context itself” on page 4. Do not use abbreviations in academic paper such as “it’s” and do not use the “it” word for syntactic expletive in academic papers. Use this concept consistently in the whole paper!

- What is used pre-trained BERT and RoBERTa models for fine-tuning process? Did authors perform a pre-training process for the BERT model or use already pre-trained models by others? The author should clearly mention this issue.

- In Tokenization and embeddings section on page 8, the paper mentioned the use of ‘bert-base-uncased’ tokenizers for tokenization. However, ‘bert-base-uncased’ and ‘roberta-base’ tokenizers are pre-trained in English. How these tokenizers can be implemented in Bengali language?

- What are M, N, and T terms in the BERT model? Please define them!

- In the BERT model section on page 10, please briefly define these unique tokens i.e., [CLS], [SEP], and [EOS]!

- If the authors want to elaborate Q, K, and V parameters, how do these parameters come and the correlation with the BERT inputs, i.e., X and Y?

- Please paraphrase this sentence “It is marked as “IsNext” if it does, and “NotNext” if it doesn’t.” What does “it” refer to?

- As we can see in Table 2 and 3, the number of each data class or category is imbalanced, how the proposed model can address this issue?

- Are Bangla and Bengali different terms? If they are the same term, please choose one of them and use consistently the selected term.

- In the performance evaluation, the proposed model used cross-validation, please provide more explanation in the implementation such as what is the value of the k parameter.

- In the experimental result of aspect detection and sentiment classification in this research, please also include the results if the model only uses TF-IDF and RF for the classification task to be presented in Tables 7-10. How is the performance of TF-IDF and RF?

- The authors can cite this paper https://doi.org/10.1186/s40537-023-00782-9 that also proposed hybrid strategy for sentiment analysis.

Reviewer #2: This paper proposes a mix of random forest and pretrained transformer model for two Aspect Based Sentiment Analysis (ABSA) tasks: aspect category detection and aspect sentiment classification. This study has been primarily targeted towards Bengali language and it used two existing Bengali text data in their experiments. The results show that the proposed tRF-BERT model outperforms ABSA tasks compared to tasks done with independent models.

Pros:

1. ABSA for low resource languages like Bengali is interesting

2. The proposed model looks like an ensemble model for ABSA tasks, which is interesting

3. Focus on two different tasks is a plus

Cons:

1. There is no novelty in this paper. The only novel part in this paper is combining results of two different classifiers and feeding them to a neural network model for final classification. However, this looks like an extended random forest algorithm and it just gets more data for a classification task.

2. There are several questions with experiments:

(a) Why the proposed model has not been evaluated against other ABSA models? This area is fast growing research and many methods can be applied approximately to other languages too.

(b) Why results of ABSA tasks just with random forest are not given?

(c) I couldn't find any citations for the dataset

(d) Why the runtime is not compared? Is this performance boost worth the execution time this model takes?

(e) Why no experiments done on English datasets?

3. The random forest model uses TF-IDF features. This totally ignores the concept of language models and makes it inefficient and unreliable for ABSA tasks. For TF-IDF features, the model should know all the data before hand, including some of the words from the test data. How this method can scale for larger problems?

4. Is it not possible to add other models like SVM, CNN, or other traditional models. How the performance change in that scenario?

5. It is evident from previous methods that random forest perform very poorly in ABSA tasks. How does it make sense to give equal importance to both model predictions? Can it be weighted?

6. I do not understand how the predictions are fed again into a neural network

Overall this paper lacks novelty and it requires significant addition of contributions to go for another round of submission.

6. PLOS authors have the option to publish the peer review history of their article (what does this mean?). If published, this will include your full peer review and any attached files.

Reviewer #1: No

Reviewer #2: No

---

## [Author Response · Author response to Decision Letter 0]

24 Apr 2024

Journal: PLOS ONE

Manuscript ID: PONE-D-24-01459

Title: tRF-BERT: A transformative approach to aspect-based sentiment analysis in the

bengali language

Authors: Shihab Ahmed, Moythry Manir Samia, Maksuda Haider Sayma, Md Mohsin

Kabir, M. F. Mridha

Dear Editor and Reviewers,

The authors thank all reviewers and the editor for their acceptance of revising our paper

and for their specific and essential comments. We have revised the paper and restructured

several sections. The updated version presents all the changes.

Response of Editor

Editor Comment-1: Given the journal’s biomedical and public health focus, some

applications of sentiment analysis in public health research should be highlighted

in the Introduction section (see: https://pubmed.ncbi.nlm.nih.gov/37376407;

https://pubmed.ncbi.nlm.nih.gov/37358808).

Author’s Response: We thank the Editor for suggesting these studies for improving our

paper.

Author’s Action: To address the editor’s concern, two new paragraphs were added at the

beginning of the “Introduction” section, along with the recommended studies as examples.

Editor Comment-2: Although the authors provided a description of the proposed

hybrid transformative Random Forest and Bidirectional Encoder Representations

from Transformers (tRF-BERT) model, there is limited explanation

on the technical underpinnings, such as the specific architecture details, the interaction

between Random Forest and BERT components, and how exactly the

hybrid model outperforms its constituent parts.

Author’s Response: We appreciate the Editor for emphasizing this crucial aspect to

enhance our paper.

Author’s Action: Architectural details of both BERT and Random Forest are provided in

the “BERT model” and “RF” subsections of the “Methods and Materials” section of the

manuscript. Also check Figs 2, 3, 4 in the manuscript. Additionally, the specifics of the

proposed tRF-BERT model are presented in the “Proposed tRF-BERT Model” subsection.

Only the newly added segments are highlighted in blue in the manuscript.

Editor Comment-3: While the manuscript mentions using two open-source

benchmark datasets, Cricket and Restaurant, for Aspect-Based Sentiment Analysis,

I am unable to find any citations for these datasets, and the authors do not

provide further statistics about these datasets (e.g., number of samples, distri-

1

bution of classes). A more thorough dataset description is necessary.

Author’s Response: We express our gratitude to the Editor for emphasizing the importance

of this aspect in enhancing our paper.

Author’s Action: The citations for the datasets and the link to the data source are included

in the “Data Availability” section of the manuscript. Additional statistical information regarding

the datasets, such as the number of samples and class distribution, has been included

in the “Data source” subsection within the “Data collection and preprocessing” section in

Tables 2 and 3 in the manuscript.

Editor Comment-4: The comparison against existing works primarily focuses

on the final performance metrics. A comparison which discusses the nature of the

datasets used in those works, model complexities, and computational resources

required, would offer a clearer picture of the proposed model’s advantages and

limitations.

Author’s Response: We appreciate the Editor’s helpful suggestion.

The comparison between related works and ours is presented in Table 14 in the manuscript.

All the studies discussed in Table 14 use the same two datasets as we did. The statistics

related to these two datasets are provided in Tables 2 and 3 in the manuscript.

Editor Comment-5: The author strongly focuses on F1 score and accuracy

for evaluating model performance. Incorporating additional metrics such as

Precision-Recall AUC, Matthews Correlation Coefficient, or analysis on the

model’s performance across different aspects/categories could provide a more

comprehensive evaluation.

Author’s Response: We thank the Editor for this observation. We are especially grateful

to the editor for suggesting additional metrics such as Precision-Recall AUC and Matthews

Correlation Coefficient, as well as recommending analysis of the model’s performance across

different aspects/categories. We intend to include these suggestions in future editions.

Author’s Action: We incorporated accuracy, precision, recall, and F1-score as evaluation

metrics, which helped us evaluate the models and compare them with others in the domain.

The response to this comment can be found in the “Performance Evaluation” subsection in

“Methods and materials” section and “Result Analysis” section of the manuscript.

Editor Comment-6: While some hyperparameters are listed, the process of

selecting these values or any optimization strategy employed is not discussed.

Detailing the hyperparameter tuning process, including the range of values explored,

would strengthen the methodological rigor.

Author’s Response: We thank the editor for pointing out this important point.

Author’s Action: We have tried our best to address this issue in our manuscript in the

subsection “Hyperparameters” of the “Methods and Materials” section. The first paragarph

has been added to the manuscript and colored blue as a response to this comment.

2

Response of Reviewer-1

Reviewer-1 Comment-1: In the abstract section, there is a sentence “it was clear

that all the models used for our work achieved better results than any of the

previous work”, what does the “it” word refer to? If it is not unclear, please

revise!

Author’s Response: We thank the reviewer for the observation.

Author’s Action: We revised the mentioned sentence in the “Abstract” section for better

understanding.

Reviewer-1 Comment-2: In the sentence “A crucial part of this research

involved finding or creating a dataset specifically designed for Bengali aspectlevel

sentiment analysis” on page 2, the authors mention that this paper creating

a dataset. However, the page 7 mentioned that this study used publicly available

datasets. Please clarify this issue!

Author’s Response: We thank the reviewer for mentioning the issue. In our research

work, we didn’t create a new dataset; instead, we used publicly available datasets.

Author’s Action: We changed the sentence mentioned in the “Introduction” section making

the matter clear.

Reviewer-1 Comment-3: There is the sentence “it’s crucial to consider the

semantics of the given aspect as a new and distinct piece of information, separate

from the context itself” on page 4. Do not use abbreviations in academic paper

such as “it’s” and do not use the “it” word for syntactic expletive in academic

papers. Use this concept consistently in the whole paper!

Author’s Response: We thank the reviewer for making this suggestion regarding our paper.

Author’s Action: We updated the mentioned sentence according to the reviewer’s advice in

the first paragraph of the “Literature Review” section.

Reviewer-1 Comment-4: What is used pre-trained BERT and RoBERTa models

for fine-tuning process? Did authors perform a pre-training process for the

BERT model or use already pre-trained models by others? The author should

clearly mention this issue.

Author’s Response: We thank the reviewer for mentioning the issue. For this study, we

used already pre-trained BERT and RoBERTa models by others.

Author’s Action: Please refer to the first paragraph with the heading “BERT-based model”

in the “Proposed tRF-BERT model” subsection in the “Methods and materials” section of

the manuscript.

Reviewer-1 Comment-5: In Tokenization and embeddings section on page 8,

the paper mentioned the use of ‘bert-base-uncased’ tokenizers for tokenization.

However, ‘bert-base-uncased’ and ‘roberta-base’ tokenizers are pre-trained in

English. How these tokenizers can be implemented in Bengali language?

3

Author’s Response: We thank the reviewer for focusing on this issue.

Author’s Action: A short explanation of using ‘bert-base-uncased’ and ‘roberta-base’ tokenizers

for our Bengali datasets is added to the manuscript. We added this explaination

in “Tokenization and embeddings” subsection of “Methods and materials” section of our

manuscript.

Reviewer-1 Comment-6: What are M, N, and T terms in the BERT model?

Please define them!

Author’s Response: We thank the reviewer for the observation.

Author’s Action: We briefly defined M, N, and T in “BERT model” sub-subsection of

“Aspect and sentiment prediction models” subsection of “Methods and materials” section.

Reviewer-1 Comment-7: In the BERT model section on page 10, please briefly

define these unique tokens i.e., [CLS], [SEP], and [EOS]!

Author’s Response: We thank the reviewer for the observation.

Author’s Action: We briefly defined the unique tokens i.e., [CLS], [SEP], and [EOS] in

“BERT model” sub-subsection of “Aspect and sentiment prediction models” subsection of

“Methods and materials” section.

Reviewer-1 Comment-8: If the authors want to elaborate Q, K, and V parameters,

how do these parameters come and the correlation with the BERT

inputs, i.e., X and Y?

Author’s Response: We thank the reviewer for the observation.

Author’s Action: We elaborated on Q, K and V parameters and their correlation with

BERT inputs in “BERT model” sub-subsection of “Aspect and sentiment prediction models”

subsection of “Methods and materials” section.

Reviewer-1 Comment-9: Please paraphrase this sentence “It is marked as

“IsNext” if it does, and “NotNext” if it doesn’t.” What does “it” refer to?

Author’s Response: We appreciate the reviewer’s helpful suggestion. The word ”it” refers

to the relationship between the second sentence and the first sentence in the original text.

Author’s Action: However, we paraphrased the sentence for better understanding in the

“BERT model” sub-subsection of the “Aspect and sentiment prediction models” subsection

of “Methods and materials” section.

Reviewer-1 Comment-10: As we can see in Table 2 and 3, the number of each

data class or category is imbalanced, how the proposed model can address this

issue?

Author’s Response: We thank the reviewer for mentioning the issue. To evaluate the

performance of our model, we employed a combination of metrics: Accuracy, Precision, Recall,

and F1-Score. While Accuracy is sufficient for balanced datasets, imbalanced datasets

require a more nuanced approach. In such cases, Precision, Recall, and F1-Score provide

valuable insights. By considering these metrics, we gain a comprehensive understanding of

how well our model identifies every case, even when the classes are not evenly distributed.

4

Author’s Action: As shown in Tables 7, 8, 9 and 10 in the manuscript, our proposed model

demonstrates strong performance across all evaluation metrics, indicating its effectiveness

in handling both balanced and imbalanced scenarios. A paragraph addressing this issue is

added in the “Results” subsection right above Table 7 of the “Result analysis” section in the

manuscript.

In future work, we aim to address the issue of imbalanced data classes by exploring various

data balancing techniques such as oversampling, undersampling, and class weighting.

Reviewer-1 Comment-11: Are Bangla and Bengali different terms? If they

are the same term, please choose one of them and use consistently the selected

term.

Author’s Response: We thank the reviewer for pointing out this important point. Bengali

is the same term as Bangla. Bangla is the language’s name in Bangla, while Bengali is the

term used in English.

Author’s Action: To avoid confusion, we used “Bengali” consistently throughout the paper.

Reviewer-1 Comment-12: In the performance evaluation, the proposed model

used cross-validation, please provide more explanation in the implementation

such as what is the value of the k parameter.

Author’s Response: We appreciate the esteemed reviewer for highlighting this concern.

Author’s Action: In response, we have included the value of K in the “Cross-Validation”

subsection of the “Methods and materials” section of the manuscript. Only the newly added

lines are highlighted in blue.

Reviewer-1 Comment-13: In the experimental result of aspect detection and

sentiment classification in this research, please also include the results if the

model only uses TF-IDF and RF for the classification task to be presented in

Tables 7-10. How is the performance of TF-IDF and RF?

Author response: We are grateful to the respected reviewer for the valuable comments.

Author’s Action: We have included the results obtained from the Random Forest model with

TF-IDF in Tables 7, 8, 9, and 10 in the manuscript.

Reviewer-1 Comment-14: The authors can cite this paper https://doi.org/10.1186/s40537-

023-00782-9 that also proposed hybrid strategy for sentiment analysis.

Author’s Response: We thank the reviewer for suggesting this supporting material for our

study. Author’s Action: We have cited this study in the seventh paragraph in “Introduction”

section of the manuscript.

5

Response of Reviewer-2

Reviewer 2 Comment-1:This paper proposes a mix of random forest and pretrained

transformer model for two Aspect Based Sentiment Analysis (ABSA)

tasks: aspect category detection and aspect sentiment classification. This study

has been primarily targeted towards Bengali language and it used two existing

Bengali text data in their experiments. The results show that the proposed tRFBERT

model outperforms ABSA tasks compared to tasks done with independent

models.

Pros: 1. ABSA for low resource languages like Bengali is interesting 2. The

proposed model looks like an ensemble model for ABSA tasks, which is interesting

3. Focus on two different tasks is a plus

Author response: We thank the respectful reviewer for these excellent remarks.

Reviewer 2 comment-2: Cons: 1. There is no novelty in this paper. The

only novel part in this paper is combining results of two different classifiers and

feeding them to a neural network model for final classification. However, this

looks like an extended random forest algorithm and it just gets more data for a

classification task.

Author’s Response: We thank the reviewer for providing such insightful observations.

We have tried our best to respond.

Author’s Action: Our proposed model outperformed all existing works on the publicly available

‘Cricket’ and ‘Restaurant’ datasets in the field of Bengali ABSA. The contributions

made by our study are the markers that ensure the novelty of the research in the field of

Bengali ABSA. The contributions made by our study are mentioned in the “Introduction”

section of the manuscript.

Reviewer 2 Comment-3:There are several questions with experiments: (a)

Why the proposed model has not been evaluated against other ABSA models?

This area is fast growing research and many methods can be applied approximately

to other languages too. (b) Why results of ABSA tasks just with random

forest are not given? (c) I couldn’t find any citations for the dataset (d) Why

the runtime is not compared? Is this performance boost worth the execution

time this model takes? (e) Why no experiments done on English datasets?

Author’s Response: We are grateful to our respected reviewer for the valuable comments.

(a) Author’s Action: Our research primarily concentrated on Bengali Aspect-Based Sentiment

Analysis (ABSA) tasks. As such, our investigation was limited to comparing our

model exclusively with existing Bengali ABSA models. The comparison with existing Bengali

ABSA models is given in Table 14 in our manuscript in the “Comparison with previous

works” subsection. However, we aim to extend our evaluation in future studies to include

comparisons with models designed for various other languages.

(b) Author’s Action: We have included the results obtained from the Random Forest

6

model with TF-IDF in Tables 7, 8, 9, and 10 in the manuscript.

(c) Author’s Action: The datasets used in this study are available at [https://github.com/atik-

05/Bangla ABSA Datasets] Additionally, citation 16 in the reference section provides further

details about the dataset and includes the DOI link.

(d) Author’s Action: We have incorporated a comparison of runtime in Table 11 in the

manuscript in the “Results” subsection in the “Result and analysis” sec

---

## [Decision Letter · Decision Letter 1]

13 May 2024

PONE-D-24-01459R1tRF-BERT: A transformative approach to aspect-based sentiment analysis in the bengali languagePLOS ONE

Dear Dr. Mridha,

Thank you for submitting your manuscript to PLOS ONE. After careful consideration, we feel that it has merit but does not fully meet PLOS ONE’s publication criteria as it currently stands. Therefore, we invite you to submit a revised version of the manuscript that addresses the points raised during the review process.

We look forward to receiving your revised manuscript.

Kind regards,

Qin Xiang Ng, MBBS, MPH

Academic Editor

PLOS ONE

Journal Requirements:

Reviewers' comments:

Reviewer's Responses to Questions

**Comments to the Author**

1. If the authors have adequately addressed your comments raised in a previous round of review and you feel that this manuscript is now acceptable for publication, you may indicate that here to bypass the “Comments to the Author” section, enter your conflict of interest statement in the “Confidential to Editor” section, and submit your "Accept" recommendation.

Reviewer #1: (No Response)

2. Is the manuscript technically sound, and do the data support the conclusions?

Reviewer #1: Yes

3. Has the statistical analysis been performed appropriately and rigorously? 

Reviewer #1: Yes

4. Have the authors made all data underlying the findings in their manuscript fully available?

Reviewer #1: Yes

5. Is the manuscript presented in an intelligible fashion and written in standard English?

Reviewer #1: Yes

6. Review Comments to the Author

Reviewer #1: The authors generally already addressed my previous comments. However, there is a point that should be clarified by the author as following:

- The authors said that they used "pre-trained BERT and RoBERTa models by others". Please give the reference where those pre-trained models can be found.

7. PLOS authors have the option to publish the peer review history of their article (what does this mean?). If published, this will include your full peer review and any attached files.

Reviewer #1: No

---

## [Author Response · Author response to Decision Letter 1]

19 May 2024

Response of Reviewer-1

Reviewer-1 Comment-1: The authors generally already addressed my previous

comments. However, there is a point that should be clarified by the author as

following: - The authors said that they used ”pre-trained BERT and RoBERTa

models by others”. Please give the reference where those pre-trained models

can be found.

Author’s Response: We thank the reviewer for pointing out the issues.

Author’s Action: We’ve already referenced two papers in Citation 14 and 32, providing

in-depth insights into pre-trained models initially, though not uniformly. Now, these references

are consistently cited throughout the text wherever pre-trained models are mentioned.

---

## [Decision Letter · Decision Letter 2]

7 Jun 2024

PONE-D-24-01459R2tRF-BERT: A transformative approach to aspect-based sentiment analysis in the bengali languagePLOS ONE

Dear Dr. Mridha,

Thank you for submitting your manuscript to PLOS ONE. After careful consideration, we feel that it has merit but does not fully meet PLOS ONE’s publication criteria as it currently stands. Therefore, we invite you to submit a revised version of the manuscript that addresses the points raised during the review process.

We look forward to receiving your revised manuscript.

Kind regards,

Qin Xiang Ng, MBBS, GDMH, MPH

Academic Editor

PLOS ONE

Journal Requirements:

Reviewers' comments:

Reviewer's Responses to Questions

**Comments to the Author**

1. If the authors have adequately addressed your comments raised in a previous round of review and you feel that this manuscript is now acceptable for publication, you may indicate that here to bypass the “Comments to the Author” section, enter your conflict of interest statement in the “Confidential to Editor” section, and submit your "Accept" recommendation.

Reviewer #1: All comments have been addressed

Reviewer #2: All comments have been addressed

2. Is the manuscript technically sound, and do the data support the conclusions?

Reviewer #1: Yes

Reviewer #2: Partly

3. Has the statistical analysis been performed appropriately and rigorously? 

Reviewer #1: Yes

Reviewer #2: N/A

4. Have the authors made all data underlying the findings in their manuscript fully available?

Reviewer #1: Yes

Reviewer #2: Yes

5. Is the manuscript presented in an intelligible fashion and written in standard English?

Reviewer #1: Yes

Reviewer #2: Yes

6. Review Comments to the Author

**Reviewer #1: **(No Response)

**Reviewer #2:** Authors have addressed most of the previous reviews. However, I have the following comments for this revised manuscript:

Use of TF-IDF features in the proposed model: TF-IDF performs decent, not only in the ABSA problem, but also for several other NLP classifiers. The only problem with TF-IDF is its lack of generalization. Authors did not show any evidence how the model performs if tokens are not present during the training phase but becomes available during the test. How about other classifier models like SVM and NB instead of Random Forest? They are traditional ML models too.

7. PLOS authors have the option to publish the peer review history of their article (what does this mean?). If published, this will include your full peer review and any attached files.

Reviewer #1: No

Reviewer #2: No

---

## [Author Response · Author response to Decision Letter 2]

10 Jul 2024

Response of Reviewer-2

Reviewer-2 Comment-1: Authors have addressed most of the previous reviews. However, I have the following comments for this revised manuscript:

Use of TF-IDF features in the proposed model: TF-IDF performs decent, not only in the ABSA problem, but also for several other NLP classifiers. The only problem with TF-IDF is its lack of generalization. Authors did not show any evidence how the model performs if tokens are not present during the training phase but becomes available during the test. How about other classifier models like SVM and NB instead of Random Forest? They are traditional ML models too.

Author’s Response: We thank the reviewer for pointing out the issues. We have tried our best to make the use of the tokenizer during the training phase more explicit.

Author’s Action: It is known that TF-IDF works well with trained words or vocabularies. In our research, we used vocabularies that are most commonly used in sentiment analysis, and as a result, TF-IDF performed satisfactorily. For future work, we intend to explore other technologies such as word2vec and BERT tokenizers to address the issue of generalization and improve the model’s performance with previously unseen tokens.

The tokens are present during the training phase, as shown in Figure 4 on page 17 (Section “Methods and Materials”). We have also added a statement indicating the use of the tok- enizer with TF-IDF in the last paragraph of the subsection “Tokenization and Embeddings” in the “Methods and Materials” section. (New additions are coloured blue.)

Additionally, we have experimented with SVM and NB instead of Random Forest with the BERT model, and the results are mentioned in Table 13 on page 22. To avoid confusion, we have updated the names of the models.

---

## [Editor Report · Decision Letter 3]

17 Jul 2024

tRF-BERT: A transformative approach to aspect-based sentiment analysis in the bengali language

PONE-D-24-01459R3

Dear Dr. Mridha,

We’re pleased to inform you that your manuscript has been judged scientifically suitable for publication and will be formally accepted for publication once it meets all outstanding technical requirements.

Kind regards,

Qin Xiang Ng, MBBS, GDMH, MPH

Academic Editor

PLOS ONE
---

## [Editor Report · Acceptance letter]

2 Aug 2024

PONE-D-24-01459R3 

PLOS ONE

Dear Dr. Mridha, 

I'm pleased to inform you that your manuscript has been deemed suitable for publication in PLOS ONE. Congratulations! Your manuscript is now being handed over to our production team.

Kind regards, 

on behalf of

Dr. Qin Xiang Ng 

Academic Editor

PLOS ONE